# Assessing fall risk and equilibrium function in patients with age-related macular degeneration and glaucoma: An observational study

Takahiro Tokunaga[1]*, Rinako Takegawa[1], Yoshiki Ueta[2], Yasuhiro Manabe[1], Hiroaki Fushiki[3]

1 Otorhinolaryngology, Shinseikai Toyama Hospital, Imizu, Japan, 2 Ophthalmology, Shinseikai Toyama Hospital, Imizu, Japan, 3 Otolaryngology, Mejiro University Ear Institute Clinic, Saitama, Japan

* t-toku3@nifty.com

**Data Availability Statement:** All relevant data are within the paper and its Supporting information files.

## Abstract

### Background

Falls in older adults are a significant public health concern, and age-related macular degeneration (AMD) and glaucoma have been identified as potential visual risk factors. This study was designed to assess equilibrium function, fall risk, and fall-related self-efficacy (an individual's belief in their capacity to act in ways necessary to reach specific goals) in patients with AMD and glaucoma.

### Methods

This observational study was performed at the Otorhinolaryngology Department of Shinseikai Toyama Hospital. The cohort comprised 60 participants (AMD; n = 30; median age, 76.0 years; and glaucoma; n = 30; median age, 64.5 years). Visual acuity and visual fields were assessed using the decimal best-corrected visual acuity and Humphrey visual field tests, respectively. The evaluation metrics included pathological eye movement analysis, bedside head impulse test, single-leg upright test, eye-tracking test, optokinetic nystagmus, and posturography. Furthermore, we administered questionnaires for fall risk determinants including the Dizziness Handicap Inventory, Activities-Specific Balance Confidence Scale, Falls Efficacy Scale-International, and Hospital Anxiety and Depression Scale. The collected data were analyzed using descriptive statistics, and Spearman's correlation analysis was employed to examine the interrelations among the equilibrium function, fall risk, and other pertinent variables.

### Results

Most participants exhibited standard outcomes in equilibrium function evaluations. Visual acuity and field deficits had a minimal impact on subjective dizziness manifestations, degree of disability, and fall-related self-efficacy. Both groups predominantly showed high self-efficacy. No significant correlation was observed between visual acuity or field deficits and

**Funding:** The author(s) received no specific funding for this work.

**Competing interests:** The authors have declared that no competing interests exist.

body equilibrium function or fall risk. However, greater peripheral visual field impairment was associated with a tendency for sensory reweighting from visual to somatosensory.

## Conclusion

Self-efficacy was higher and fall risk was relatively lower among patients with mild-to-moderate visual impairment, with a tendency for sensory reweighting from visual to somatosensory in those with greater peripheral visual field impairment. Further studies are required to validate these findings.

## Introduction

Falls are a pressing public health issue, particularly among older adults. Data from the Centers for Disease Control and Prevention in 2018 indicated that 27.5% of adults aged $\geq$65 years in the United States had experienced at least one fall in the preceding year, with 10.2% sustaining injuries due to these falls [1]. The risk of falls in older adults is caused by a complex combination of factors. For example, as older adults spend most of their time at home, fall risk factors in the home environment, including inadequate lighting, slippery surfaces, worn carpets, and stairs without handrails, become important. Additionally, visual deficits, which are prevalent in older adults, have been identified as potential risk factors of falls.

The integration of visual, vestibular, and somatosensory inputs is essential for postural regulation. Any deficit in these sensory modalities can jeopardize postural steadiness and increase the fall risk [2]. Although visual deficits are not the only cause of falls, discerning their specific risk factors can facilitate the formulation of robust preventive measures.

Previous studies have provided insights into the relationships among visual acuity, field deficits, and the incidence of falls. For instance, one study revealed that first-eye cataract surgery significantly reduces the rate of falls and enhances overall visual function and health status [3]. Another study suggested that while baseline bilateral visual impairment was associated with a nearly two-fold increase in the risk of falls, even mild unilateral visual impairment was significantly associated with frequent falls due to both correctable and uncorrectable conditions [4]. Additionally, a cohort study highlighted that recently developed visual impairments in older adults increased the likelihood of subsequent falls and fractures within a 5-year period [5]. In contrast, research focusing on patients with glaucoma found that deficits in postural control, although affected by visual field deficit severity, were not solely attributable to impaired peripheral visual input [6].

The exact correlation between visual deficits and falls remains unclear and, at times, is contradictory. The precise risk factors that lead to falls in individuals with visual challenges are not comprehensively understood because of research limitations. This knowledge gap impedes the creation of specialized fall prevention strategies for the visually impaired.

In this study, we included patients with age-related macular degeneration (AMD) and glaucoma with moderate or low severity to distinguish between visual acuity impairment and visual field impairment, and determine which of the two contributes to postural control. AMD is a degenerative condition that affects the macula and the central region of the retina, leading to visual acuity impairment. Globally, AMD causes irreversible visual impairment, which is sometimes bilateral and can lead to blindness in severe cases [7]. As the Japanese population ages, the number of AMD cases has surged. Glaucoma, which is characterized by optic nerve damage primarily due to elevated intraocular pressure, results in visual field narrowing and

loss. Over 10% of individuals aged ≥60 years in Japan have glaucoma, making it the primary cause of blindness in the country [8].

This study examined the equilibrium function and fall susceptibility in patients with AMD and glaucoma attending a standard ophthalmology outpatient clinic without subjective vertigo symptoms. By measuring the fall risk in a population of mild–to-moderately visually impaired individuals, this research may help develop fall prevention strategies for such individuals.

## Materials and methods

### Study design and participants

This observational study was conducted at the Otorhinolaryngology Department of Shinseikai Toyama Hospital between April 1, 2021, and March 31, 2022. Patients diagnosed with AMD and glaucoma, representing visual acuity and visual field deficits, respectively, were enrolled from the Ophthalmology Department of our hospital. Thirty patients with AMD (median age, 76.0 years) and 30 patients with glaucoma (median age, 64.5 years) participated in this study (Table 1). All patients were current attendees of the hospital, and their medical records were used to investigate their medical history and cognitive function. The mean left–right difference in hearing of all study participants was <15 dB. Exclusion parameters included any neurological or musculoskeletal alignment history that could affect balance and gait, potentially confounding the visual impairment-related fall risk. Patients with a history of vertigo or a diagnosis of vestibular dysfunction by a specialist were excluded. Patients diagnosed with cognitive impairment were excluded because cognitive impairment can affect the results of the questionnaire.

### Visual function assessment

Monocular visual acuity was assessed in patients with AMD using decimal best-corrected visual acuity (BCVA) with a Landolt ring. In contrast, the monocular visual fields of glaucoma patients were assessed using the MD value (dB) from the Humphrey visual field test. BCVA

**Table 1. Profile of the participants.**

|  |  | Age-related macular degeneration | Glaucoma |
|---|---|---|---|
|  |  | N = 30 | N = 30 |
| Age |  | 76.0 [70.0 to 79.0] | 64.5 [58.0 to 69.0] |
| Sex | Female | 6 (20%) | 15 (50%) |
|  | Male | 24 (80%) | 15 (50%) |
| Decimal best-corrected visual acuity | Right | 0.75 [0.3 to 1.2] | 1.2 [1.2 to 1.5] |
|  | Left | 0.95 [0.8 to 1.2] | 1.2 [1.0 to 1.2] |
| MD value (dB) of the Humphrey visual field test | Right | - | −2.70 [−5.21 to −0.67] |
|  | Left | - | −2.37 [−5.38 to −1.09] |
| Complications |  |  |  |
| Dementia |  | 0 (0%) | 0 (0%) |
| Diabetes mellitus |  | 6 (20%) | 6 (20%) |
| Dyslipidemia |  | 10 (33%) | 7 (23%) |
| Hypertension |  | 14 (47%) | 8 (27%) |
| Arrhythmia |  | 1 (3%) | 2 (7%) |
| Other heart diseases |  | 2 (7%) | 4 (13%) |
| Osteoporosis |  | 1 (3%) | 0 (0%) |

were transmuted to "logMAR" for arithmetic computations and statistical analysis.

$$logMAR = \log_{10}\left(\frac{1}{Decimal\ BCVA}\right)$$

High logMAR values signify low visual acuity, whereas low MD values indicate a compromised visual field. The superior-performing eye's values were utilized for both visual acuity and visual field evaluations. Humans are semi-crossed binocular animals; therefore, they rely on information from the better eye to discriminate objects and control eye movements and postural reflexes derived from peripheral vision. Therefore, the better eye was the main object of the analysis.

We also measured the visual fields extensively using Goldmann visual field meters; however, we did not analyze these measurements because they are not routinely performed in clinical practice and may not reflect the visual fields at the time of enrollment in this study.

## Outcome measures

The outcome measures comprised the equilibrium function and fall risk evaluations. Equilibrium function evaluation included pathological eye movement analysis, bedside head impulse test (HIT), single-leg upright test, eye-tracking test (ETT), optokinetic nystagmus (OKN), and posturography. The HIT is a technique used to evaluate semicircular canal dysfunction by observing the vestibulo–ocular reflex. The examiner sat facing the participants and asked them to look at the tip of the examiner's nose. The examiner then firmly grasped the participant's temporal region with both hands and applied a fast, small rotation of the head for impulse stimulation. In cases of semicircular canal dysfunction, head impulse stimulation in the direction of the affected side produces a "catch up saccade." The head impulse was applied three times each to the left and right sides, and a positive result was obtained when the catch-up saccade was observed two or more times [9]. The ETT measures eye movements by fixing the participant's head and having the eyes tracking a smooth-moving optotype (amplitude, 40˚; frequency, 0.3 Hz) in front of the eye. Smooth eye movements were considered normal, and saccadic and ataxic patterns were considered abnormal [10]. The OKN measures eye movement while looking at an object moving at a constant angular velocity in front of the eye. Stimuli of 60˚/s were applied in the left and right directions. Saccadic patterns were considered abnormal [11]. Eye movements were recorded using the yVOG (Daiichi Medical Co., Ltd., Tokyo, Japan) and Gravicoda GW-31 (Anima Co., Ltd., Tokyo, Japan) facilitated posturography. Posturography was performed on solid or rubber foam surfaces while observing a small spot in front of a white wall and closing the eye with an eye mask. The foam ratio (posturography with/without foam) with eyes closed served as a somatosensory-dependent postural control metric, whereas the Romberg foam ratio served as a visual-dependent metric. The quotient of (Romberg ratio on foam)/(foam ratio with closed eyes) was computed as a visual postural control dependence index, named the "visual/somatosensory ratio" [12].

Patients completed the Dizziness Handicap Inventory (DHI) [13], Activities-specific Balance Confidence (ABC) scale [14], Falls Efficacy Scale-International (FES-I) [15], and the Hospital Anxiety and Depression Scale (HADS) [16]. They also documented falls over the preceding six months. The DHI is designed to quantify the magnitude of subjective physical, emotional, and functional impairments. It measures the extent of impairment in daily activities caused by dizziness. Scores range between 0 and 100 points, with categorizations of mild (0–30 points), moderate (31–60 points), and severe (60–100 points) to denote severity [17]. The ABC scale employs a 10-point scale ranging 0–100% across 16 activities of daily living. An average score of 67% or lower across these 16 items indicates an increased fall risk [18]. The

FES-I is a questionnaire focused on the fear of falling. It quantifies self-efficacy related to falls. Self-efficacy is an individual's belief in their capacity to act in the manner necessary to reach specific goals [19]. Participants rate their level of caution to avoid falling during 16 indoor and outdoor activities on a four-point scale. Cumulatively, higher scores signify lower self-efficacy regarding falls. Moreira et al. established 23 points a threshold score [20]. The HADS is a straightforward, 14-item questionnaire designed to evaluate anxiety and depression. Subscales of ≥12 for either anxiety or depression are reported to have a sensitivity of 92% and a specificity of 90% in diagnosing psychiatric morbidity [21]. All fall risk evaluations were conducted using self-report questionnaires.

### Ethical considerations

This study was approved by the Research Ethics Committee of the Shinseikai Toyama Hospital (Approval No.: 210309–1). All participants provided written informed consent and the study strictly adhered to the tenets of the Declaration of Helsinki.

### Statistical analysis

Descriptive statistical methods were used for data analysis. Categorical variables are presented as frequencies and percentages, whereas continuous variables are presented as medians and interquartile ranges (IQR). Spearman's correlation analysis examined the correlations among equilibrium function, fall risk, and other variables.

Although this research was preliminary and did not include validation testing, the sample size was deduced from previous studies. In one study, approximately 30% of patients with vestibular dysfunction had an ABC scale score of <67%, indicating an elevated fall risk [22]. Assuming a marginally reduced fall risk (20%) in our study cohort, a minimum sample size of 44 patients was deduced to achieve a 95% confidence interval (CI) ranging from 8% to 32% (60% relative accuracy). Analyses were performed using Stata software v18.0 (StataCorp LP, Texas, USA). Statistical significance was set at $P < 0.05$. 3.

## Results

### Demographics and visual impairment characteristics

The cohort included 30 patients with AMD and 30 patients with glaucoma, with females constituting 35% of the total. The participants' ages ranged from 40 to 84 years, with a median age of 70.0 years. Table 1 shows the demographic details and the extent of visual deficits among the participants.

### Evaluation of fall risk and self-efficacy pertaining to falls

In the previous six months, 7% of patients with AMD and glaucoma reported falls. The median FES-I scores were comparable between patients with AMD (23.0, IQR: 20.0 to 29.0) and glaucoma (23.5, IQR: 20.0 to 27.0). Similarly, the median ABC scale scores for AMD (95.3, IQR: 83.8 to 98.8) and glaucoma (96.3, IQR: 86.9 to 98.8) patients showed no significant difference (p = 0.76). Regarding fall-related self-efficacy, 3.3% of patients with AMD and 10.0% of patients with glaucoma registered an ABC scale score of <67%, a threshold indicative of an increased fall risk in older individuals [22]. In the single-leg upright test, 20% of patients with AMD and 3% of patients with glaucoma maintained their stance for <5 s, a duration associated with an elevated fall risk [23]. However, this difference was not statistically significant (p = 0.103).

## Oculomotor and vestibular function assessment

Table 2 shows the results of the balance function tests and fall-related self-efficacy metrics. Most participants in both groups demonstrated normal HIT, ETT, and OKN results. Specifically, 80% of the patients with AMD and 73% of the patients with glaucoma had standard ETT results. All patients with AMD and 90–97% of patients with glaucoma exhibited normal OKN results. Positive HIT results were observed in 10–13% of patients with AMD and 3% of those with glaucoma. In terms of subjective dizziness and associated disability, only a single glaucoma patient registered a DHI score ≥60.

## Balance and its association with visual deficits

Fig 1 shows the relationship between visual impairments (visual acuity and visual field deficits) and the visual/somatosensory ratio. Neither AMD nor glaucoma patients exhibited significant correlations with Spearman's correlation analysis. Among patients with glaucoma, more

**Table 2. Comparison of equilibrium function and fall risk.**

|  |  | Age-related macular degeneration | Glaucoma | p-value |
|---|---|---|---|---|
|  |  | N = 30 | N = 30 |  |
| The number of falls within the last 6 months | 0 | 28 (93%) | 28 (93%) | 1.00 |
|  | ≥1 | 2 (7%) | 2 (7%) |  |
| Single-leg upright test (second) | Right | 30.0 [5.0–30.0] | 30.0 [13.0–30.0] | 0.48 |
|  | Left | 30.0 [13.0–30.0] | 27.0 [14.0–30.0] | 0.59 |
| FES-I |  | 23.0 [20.0–29.0] | 23.5 [20.0–27.0] | 0.76 |
| DHI | Physical | 2.0 [0.0–4.0] | 0.0 [0.0–2.0] | 0.20 |
|  | Emotional | 0.0 [0.0–2.0] | 0.0 [0.0–0.0] | 0.27 |
|  | Functional | 1.0 [0.0–6.0] | 0.0 [0.0–0.0] | 0.02 |
|  | Total | 5.0 [0.0–12.0] | 0.0 [0.0–4.0] | 0.03 |
| ABC scale |  | 95.3 [83.8–98.8] | 96.3 [86.9–98.8] | 0.76 |
| HADS | Anxiety | 5.5 [3.0–7.0] | 5.0 [4.0–7.0] | 0.97 |
|  | Depression | 6.0 [4.0–8.0] | 6.0 [5.0–8.0] | 0.79 |
| HIT positive | Right | 4 (13%) | 1 (3%) | 0.17 |
|  | Left | 3 (10%) | 1 (3%) | 0.32 |
| ETT | Normal | 24 (80%) | 22 (73%) | 0.56 |
|  | Saccadic | 6 (20%) | 7 (23%) |  |
|  | Ataxic | 0 (0%) | 1 (3%) |  |
| OKN: Left | Normal | 29 (100%) | 26 (90%) | 0.08 |
|  | Saccadic | 0 (0%) | 3 (10%) |  |
| OKN: Right | Normal | 29 (100%) | 28 (97%) | 0.31 |
|  | Saccadic | 0 (0%) | 1 (3%) |  |
| Stabilometry |  |  |  |  |
| Romberg ratio on foam |  | 1.82 [1.58 to 2.17] | 1.65 [1.38 to 1.99] | 0.13 |
| The form ratio with closing eyes |  | 1.88 [1.53 to 2.54] | 1.96 [1.65 to 2.39] | 0.66 |
| Area of sway with opening eyes † |  | 2 (7%) | 4 (13%) | 0.39 |
| Area of sway with closing eyes † |  | 1 (3%) | 2 (7%) | 0.55 |
| L/A with opening eyes † |  | 1 (3%) | 3 (10%) | 0.30 |
| L/A with closing eyes † |  | 2 (7%) | 3 (10%) | 0.64 |

L/A: locus length per unit area

† shows the number of outliers.

(A)

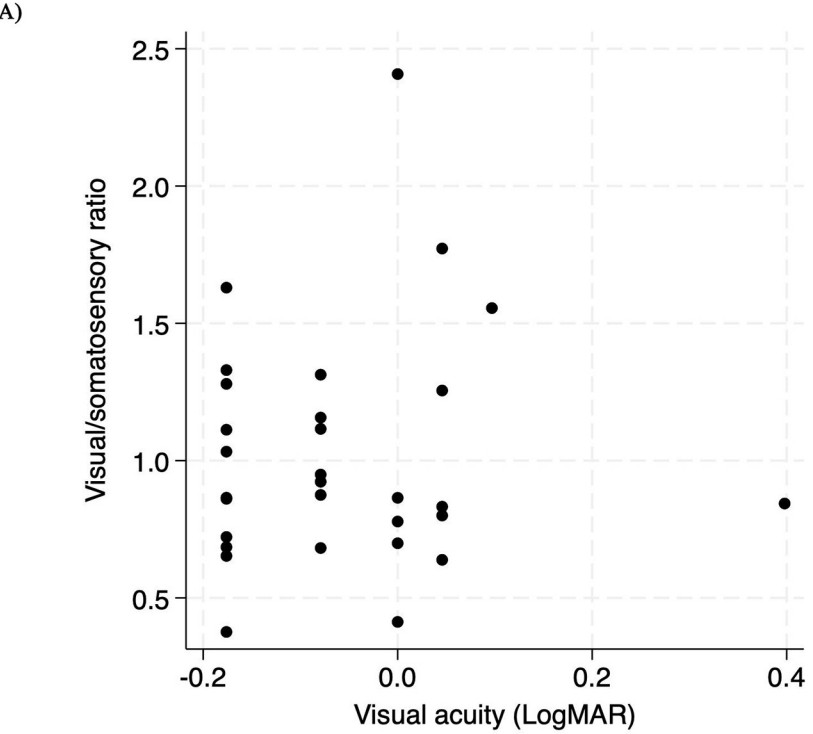

(B)

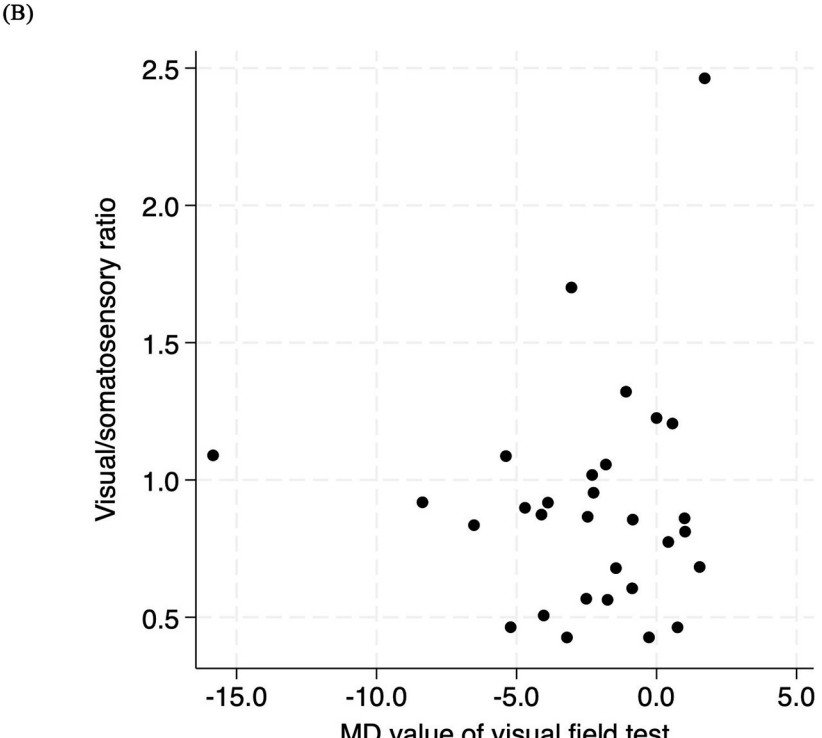

**Fig 1. Association between visual impairment and body equilibrium.** (A) Relationship between visual acuity and the visual/somatosensory ratio, (B) Relationship between visual field impairment and the visual/somatosensory ratio, The superior-performing eye's results were utilized for both visual acuity and visual field evaluations.

**Table 3. Correlation between visual deficits and equilibrium function or fall risk.**

| | | Visual acuity (LogMAR) | Visual field (MD value) |
|---|---|---|---|
| FES-I | | −0.16 | 0.02 |
| DHI | | 0.00 | 0.06 |
| ABC scale | | −0.38 | −0.06 |
| HADS | Anxiety | −0.36 | −0.13 |
| | Depression | 0.01 | 0.01 |
| Stabilometry | | | |
| Romberg ratio on foam | | −.13 | 0.16 |
| The form ratio with closing eyes | | −.15 | 0.14 |
| Area of sway with opening eyes | | 0.07 | −.23 |
| Area of sway with closing eyes | | −.06 | 0.08 |
| L/A with opening eyes | | 0.23 | −.39 |
| L/A with closing eyes | | 0.22 | −.26 |

The values represent correlation coefficients. The superior-performing eye's results were utilized for both visual acuity and visual field evaluations.

L/A: locus length per unit area

pronounced visual field impairment in the better-performing eye corresponded to reduced visual dependence. No association was identified between visual deficits and body equilibrium function or fall susceptibility across the other metrics (Table 3).

## Discussion

Our investigation aimed to evaluate the equilibrium function and susceptibility to falls in patients with AMD and glaucoma. The patients exhibited standard outcomes in the HIT, ETT, and OKN evaluations. The extent of visual acuity and visual field deficits minimally influenced subjective dizziness, degree of disability, and fall-related self-efficacy. Patients with AMD as well as those with glaucoma generally demonstrated high self-efficacy. Notably, the proportion of patients with a heightened fall risk was lower than that of patients with peripheral vestibular disorders. Of the 52 patients with peripheral vestibular disorders, 18 (34.6%) were assessed as being at risk for falls using the ABC scale [22]. This indicates that individuals with mild-to-moderate visual acuity and visual field deficits with normal vestibular and somatic balance functions might exhibit fewer dizziness manifestations and potentially a diminished fall risk compared to those with vestibular deficits.

The triad of visual, vestibular, and somatosensory inputs is crucial for postural regulation. While visual anomalies can compromise postural stability, existing literature suggests potential compensatory mechanisms via other sensory modalities [24]. Individuals with visual deficits often exhibit irregular postural reflexes and motor patterns, resulting in asymmetric muscle distribution. Nevertheless, they can acclimatize to these anomalies by harnessing non-visual feedback mechanisms such as auditory cues.

The visual/somatosensory ratio was used as an indicator of visual dependence of postural control. This ratio decreased in patients with impaired visual fields. This indicates potential sensory reweighting in patients with compromised visual fields, suggesting that other senses might compensate for the deteriorated visual fields. In contrast, these patterns were absent in patients with visual acuity deficits. Because visual acuity measures central vision, it may not be relevant to postural control. The peripheral retina is primarily responsible for the induction of

self-motion perception [25, 26]. Most studies have assigned the most important role in postural control to the peripheral retina [27].

The existing research on equilibrium function and postural stability in visually impaired individuals is equivocal. For instance, one study reported a correlation between glaucoma severity and equilibrium, noting that pronounced visual field loss in the better eye, but not in the worse eye, correlated with augmented standing sway rates. This sway persists even with eyes closed, suggesting that postural control deficits are not solely attributable to compromised peripheral visual input [6]. Another study highlighted that visually impaired individuals predominantly employ hip-centric strategies to maintain postural stability, indicating that visual deficits adversely affect postural stability [28].

The ramifications of visual deficits on postural control are contingent on the nature (congenital vs. acquired) and severity of the impairment. Acquired visual deficits tend to have a more pronounced effect on postural control than congenital ones [29]. Furthermore, an exploration of visually impaired athletes across diverse sports revealed that postural control disparities were linked to the extent of visual loss and sports-specific modalities [30]. The cohort in our study predominantly comprised individuals with mild-to-moderate acquired visual deficits. In this demographic group, the influence of visual impairment on the fall risk was minimal.

Nevertheless, this did not reduce the importance of balance interventions in this cohort. Visual acuity and field deficits may develop gradually without the patient being aware. Four cardinal interventions—education, medical assessment, physical exercise, and environmental modifications—have been identified for fall prevention among visually impaired older individuals [31]. Further empirical investigations are required to ascertain the necessity for, and evaluate the efficacy of, specialized interdisciplinary fall prevention initiatives. Vestibular stimulation rehabilitation reportedly improves the postural stability of visually impaired individuals, bringing it at par with that of individuals with normal vision [32]. A systematic review showed that a minimum of six weeks of balance and core stability training could beneficially influence fall risk in visually impaired individuals, irrespective of age, sex, or visual impairment severity [33]. Hence, exercises that stimulate the vestibular system through head and body movements are recommended to improve balance in visually impaired individuals.

The robustness of this study derives from the inclusion of patients with mild-to-moderate visual acuity and visual field deficits, particularly those with acquired conditions such as AMD and glaucoma, who might be oblivious to their gradual visual deterioration. Determining whether such latent visual deficits augment fall risk is pivotal for falls prevention in older individuals. Subjective symptoms were juxtaposed with empirical findings by combining subjective questionnaire surveys and objective balance function tests.

However, this study had some limitations. Age matching and controlling for other characteristics that could potentially influence the outcomes was not performed. This may have introduced confounding factors. Owing to its observational nature, there may have been a selection bias in patient recruitment. Specifically, the patients were asked to participate as volunteers, which may have discouraged the recruitment of patients with severe visual impairment. The study design inherently precludes causal inferences. Moreover, this study did not account for other potential postural control and fall risk factors such as muscle strength, walking speed, and environmental hazards. The relatively modest sample size may limit the generalizability of the findings.

In summary, our findings underscore that most patients with AMD and glaucoma attending an ophthalmology clinic exhibit standard postural control and have a lower fall risk than patients with vestibular disorders. This indicates that visual deficits did not substantially undermine postural stability in this cohort. Nonetheless, medical practitioners and

policymakers should consider vestibular function evaluations in visually impaired patients with a heightened fall risk due to other underlying conditions. Prospective studies with matched cohorts and large sample sizes are required to corroborate our findings and elucidate the underlying mechanisms.

## Supporting information

**S1 Data.**
(XLSX)

## Acknowledgments

The authors extend their gratitude to the certified orthoptists of Shinseikai Toyama Hospital for patient recruitment, and to the Department of Otolaryngology staff and clinical laboratory technicians of Shinseikai Toyama Hospital for their assistance with the body balance function tests. We would like to thank Honyaku Center Inc. for English language editing. ChatGPT was used only for syntax proofreading and checking spelling errors and grammar.

## Author Contributions

**Conceptualization:** Takahiro Tokunaga, Yoshiki Ueta, Yasuhiro Manabe, Hiroaki Fushiki.

**Data curation:** Takahiro Tokunaga, Rinako Takegawa.

**Formal analysis:** Takahiro Tokunaga.

**Investigation:** Takahiro Tokunaga, Rinako Takegawa.

**Methodology:** Takahiro Tokunaga, Yoshiki Ueta, Yasuhiro Manabe, Hiroaki Fushiki.

**Project administration:** Yasuhiro Manabe.

**Resources:** Takahiro Tokunaga, Rinako Takegawa, Yoshiki Ueta, Yasuhiro Manabe, Hiroaki Fushiki.

**Software:** Takahiro Tokunaga.

**Supervision:** Hiroaki Fushiki.

**Validation:** Takahiro Tokunaga, Yoshiki Ueta, Hiroaki Fushiki.

**Visualization:** Takahiro Tokunaga.

**Writing – original draft:** Takahiro Tokunaga.

**Writing – review & editing:** Takahiro Tokunaga, Rinako Takegawa, Yoshiki Ueta, Yasuhiro Manabe, Hiroaki Fushiki.

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
