## [Decision Letter · Decision Letter 0]

15 Jan 2024

PONE-D-23-39304Assessing fall risk and equilibrium function in patients with age-related macular degeneration and glaucoma: An observational studyPLOS ONE

Dear Dr. Tokunaga,

Thank you for submitting your manuscript to PLOS ONE. After careful consideration, we feel that it has merit but does not fully meet PLOS ONE’s publication criteria as it currently stands. Therefore, we invite you to submit a revised version of the manuscript that addresses the points raised during the review process.

We look forward to receiving your revised manuscript.

Kind regards,

Renato S. Melo, PhD

Academic Editor

PLOS ONE

Journal Requirements:

Additional Editor Comments:

Dear authors, The reviewers have reviewed the manuscript and the majority of them have decided that the manuscript has merit, however, it still needs adjustments before it can be considered for publication. Therefore, I decided to reconsider the manuscript after major revisions.

Reviewers' comments:

Reviewer's Responses to Questions

**Comments to the Author**

1. Is the manuscript technically sound, and do the data support the conclusions?

Reviewer #1: Partly

Reviewer #2: Yes

Reviewer #3: Yes

2. Has the statistical analysis been performed appropriately and rigorously? 

Reviewer #1: Yes

Reviewer #2: Yes

Reviewer #3: No

3. Have the authors made all data underlying the findings in their manuscript fully available?

Reviewer #1: Yes

Reviewer #2: Yes

Reviewer #3: Yes

4. Is the manuscript presented in an intelligible fashion and written in standard English?

Reviewer #1: Yes

Reviewer #2: Yes

Reviewer #3: Yes

5. Review Comments to the Author

Reviewer #1: Fall risk

24, 30 What is self-efficacy?

62 “deficits in postural control”

63 Make the purpose of the paper clearer. Are you attempting to replicate results of [8]? You don’t give any information about why or what about it is deficient. Recommend removing everything from line 63 to end of paragraph, replacing with some sort of justification for the research. Please also include why AMD is being included, as it has not been previously mentioned in the introduction.

82 how did you know what their vestibular function was?

114 what was the optical surround during the test?

141-149 Remove this, as it is already demonstrated in the table.

153 Within six months previously?

152 define self-efficacy here, and in abstract

184 Not sure of the justification for using superior eye. What would the results be if you used the inferior performing eye, which after all will be providing peripheral vision in half the environment.

199 inferior? Maybe you mean less. I’m not sure why a comparison to vestibular-deficient patients is given here, as it isn’t a major point of the paper

200 dizziness/vertigo is not a sensitive or specific symptom for vestibular loss. Please clarify.

201 indicates not insinuates—different meaning

212 The change in ratio is entirely expected—the term “intriguing” would indicate the opposite

215 indicate that “visual acuity” will be a measure of central vision (some readers might not know)

220 did they test the other eye?

229 “cohort” or “population” as a group, not “participants,” served to “comprise” individuals.

235 “identified” not “earmarked”

259 In 231 you indicated “the influence of visual impairment on fall risk appeared minimal” but here it actually decreases the risk? Are you comparing to effect of vestibular loss?

Reviewer #2: In this study the parameters of fall and equilibre were determined in patients with AMD and glaucoma. The visual function parameters included only visual acuity and visual field. It is necessary to extend the correlations to additional parameters including for AMD low-luminance visual acuity and dark adaptation, and for glaucoma the peripheral visual field limits as determined by Humphrey suprathershold perimetry and Goldmann perimetry.

Reviewer #3: General Comments:

This is an interesting work and I thoroughly enjoyed reading the manuscript. The authors in their manuscript have explored equilibrium function and falls risk in cohorts of age-related macular degeneration (AMD) and glaucoma. I believe that the information gained from this paper would add to the literature and knowledge base regarding falls risk in this clinical population. However, there are some issues, particularly in Method sections that need to be sufficiently addressed to improve the manuscript.

Importantly, I note that the authors have used several complex and peculiar words throughout the manuscript, for example, orchestrated, amassed, afflicted, chronicled, unbeknownst, emanates etc. I think authors have used ChatGPT, which they have acknowledged. It would be better to use simple academic words that are used in scientific writing so that it is easier for all readers to understand the contents.

Specific Comments:

Abstract:

1.The age of patients with AMD and glaucoma is missing in the Abstract. Please add this detail here.

2.Line 23:

While authors have mentioned falls determinants tools but the list of equilibrium function tests are not mentioned.

3.The authors have described 4 falls determinants tools in method sections in the main text. However, Hospital Anxiety and Depression Scale is missed in the Abstract.

4.Line 31:

Do visual deficits include both visual acuity (VA) and visual fields? Please specify this for clarity.

Introduction:

5.The introduction section would benefit by adding some further information about a wide range of falls risk factors and then stating that impaired vision is one of the important falls risk factors. For example, visual risk factors for falls in community-dwelling older adults or fall hazards in the home environment such as inadequate lighting, slippery surfaces, worn carpets, staircases without railings etc. as older adults spent the majority of their time in their homes.

6.Line 43 to 44:

It seems jumping directly to AMD and glaucoma. It would be better if the authors, in the first stance, mention that AMD and glaucoma cause central vision impairment and peripheral field losses and then describe these two conditions.

7.Line 45:

Is AMD cause of blindness or cause of irreversible visual impairment? Please check this.

8.Line 54: What does visual deficits refer to? Is it reduced vision (VA, contrast sensitivity or visual field) in general or specifically reduced VA and loss of visual fields? Please clarify this.

Methods:

9.The authors have directly reported demographic findings in the Result section. It is better to include at least the number and age of participants in this section also.

10.Line 78:

Again, it would be nice if VA and visual fields are stated clearly.

11.Visual function test:

More detail is required about vision tests.

I.What chart was used to test VA and how the process was entailed? Was VA tested monocularly or binocularly and which eye VA was used for correlation analysis in Table 3, is it better eye? I note authors have reported RE and LE in Result section in Table 1 and mentioned in the result section that better eye was used.

II.The authors have mentioned that decimal VA was converted into logMAR. I am bit curious why VA was not directly measured and recorded in logMAR.

III.It is a bit strange that authors have not tested VA in patients with glaucoma. Please justify this.

IV.How were visual fields tested monocularly or binocularly and which eye MD was used for correlation analysis, is it again better eye? Please explain these details. The authors have reported MD value for RE only in Table 1.

V.The authors have described how MD values were transmuted to power value, which is new term. Please add references to support this.

12.Outcome measures

I.It would be better to explain equilibrium function test at first and then falls risk evaluation tools later. More detailed explanation is also required for a range of equilibrium function tests.

II.The tool used by authors contain a set of questionnaires. How were responses from participants obtained? Was it self-reported or face-to-face interview? These all details are required.

13.The authors have not mentioned anything about screening of cognitive function (E.g., Mini-mental state examination) in older adults. I assume some older adults may have cognitive impairments which is likely to have impact on subjective responses that were obtained from questionnaires of falls evaluation tool. Please explain this for further clarification.

Results:

14.Line 144 to145:

The authors have directly reported results of general health but have not described in method sections that how this information was obtained.

15.In Table 2, I would suggest combining 1, 2 and 3 falls because of the smaller number of participants in each category. There are only 2 participants in each group who had falls within last 6 months. Please run this analysis again.

16.Are values reported in the Table 3 Pearson correlations? The interpretation of findings is hard to follow here.

Discussion:

This section is well written. Study limitations are well described.

17.Line 199: Please replace word inferior by less for appropriate word.

6. PLOS authors have the option to publish the peer review history of their article (what does this mean?). If published, this will include your full peer review and any attached files.

Reviewer #1: No

Reviewer #2: No

Reviewer #3: No

---

## [Author Response · Author response to Decision Letter 0]

27 Feb 2024

We would like to express our heartfelt gratitude for your review of our manuscript. We have carefully revised the manuscript and agree with all your suggestions.

Reviewer #1

Thank you for your suggestions and comments.

Line 24, 30 What is self-efficacy?

Line 152 define self-efficacy here, and in abstract

- We have added the following sentences (Line 17-19): 

This study was designed to assess equilibrium function, fall risk, and fall-related self-efficacy (an individual's belief in their capacity to act in ways necessary to reach specific goals) in patients with AMD and glaucoma.

- We have added the following sentence (Line 143-144): 

Self-efficacy is an individual's belief in their capacity to act in the manner necessary to reach specific goals.

Line 62 “deficits in postural control”

- We have revised the text as directed by Reviewer #1 (Line 63).

Line 63 Make the purpose of the paper clearer. Are you attempting to replicate results of [8]? You don’t give any information about why or what about it is deficient. Recommend removing everything from line 63 to end of paragraph, replacing with some sort of justification for the research. Please also include why AMD is being included, as it has not been previously mentioned in the introduction.

- We have changed the sentences as follows (Line 65-81): 

The exact correlation between visual deficits and falls remains unclear and, at times, is contradictory. The precise risk factors that lead to falls in individuals with visual challenges are not comprehensively understood because of research limitations. This knowledge gap impedes the creation of specialized fall prevention strategies for the visually impaired. 

In this study, we included patients with age-related macular degeneration (AMD) and glaucoma with moderate or low severity to distinguish between visual acuity impairment and visual field impairment, and determine which of the two contributes to postural control. AMD is a degenerative condition that affects the macula and the central region of the retina, leading to visual acuity impairment. Globally, AMD causes irreversible visual impairment, which is sometimes bilateral and can lead to blindness in severe cases[7]. As the Japanese population ages, the number of AMD cases has surged. Glaucoma, which is characterized by optic nerve damage primarily due to elevated intraocular pressure, results in visual field narrowing and loss. Over 10% of individuals aged ≥60 years in Japan have glaucoma, making it the primary cause of blindness in the country[8]. 

This study examined the equilibrium function and fall susceptibility in patients with AMD and glaucoma attending a standard ophthalmology outpatient clinic without subjective vertigo symptoms. By measuring the fall risk in a population of mild–moderately visually impaired individuals, this study may help develop fall prevention strategies for such individuals.

Line 82 how did you know what their vestibular function was?

- The description was ambiguous and has been corrected as follows (line 93-94): 

Patients with a history of vertigo or a diagnosis of vestibular dysfunction by a specialist were excluded.

114 what was the optical surround during the test?

- We have changed the sentence as follows (Line 128-129):

Posturography was performed on solid or rubber foam surfaces while observing a small spot in front of a white wall and closing the eye with an eye mask. 

141-149 Remove this, as it is already demonstrated in the table.

- We have removed the sentences as directed by Reviewer #1.

153 Within six months previously?

- We have changed the sentence as follows (Line 176): 

In the previous six months, 7% of patients with AMD and glaucoma reported falls.

184 Not sure of the justification for using superior eye. What would the results be if you used the inferior performing eye, which after all will be providing peripheral vision in half the environment.

- Humans are semi-crossed binocular animals; therefore, they rely on information from the better eye to discriminate objects and control eye movements and postural reflexes derived from peripheral vision. Therefore, the better eye was the main object of the analysis. This has been added to line 104-106.

199 inferior? Maybe you mean less. I’m not sure why a comparison to vestibular-deficient patients is given here, as it isn’t a major point of the paper

- We considered the fact that the fall risk for patients with mild-to moderate visual impairment was less severe than that for patients with vestibular impairment to be an important finding. We have changed the sentence as follows (Line 220-221): 

Notably, the proportion of patients with a heightened fall risk was lower than that of patients with peripheral vestibular disorders.

200 dizziness/vertigo is not a sensitive or specific symptom for vestibular loss. Please clarify.

- The description was ambiguous and has been corrected as follows (Line 221-223): 

Of the 52 patients with peripheral vestibular disorders, 18 (34.6%) were assessed as being at risk of falls using the ABC scale[22].

201 indicates not insinuates—different meaning

- We have changed the sentence as follows (line 223-): 

This indicates that individuals with mild-to-moderate visual acuity and visual field deficits...

212 The change in ratio is entirely expected—the term “intriguing” would indicate the opposite

- We have deleted the word "Intriguingly".

215 indicate that “visual acuity” will be a measure of central vision (some readers might not know)

- Thank you for the suggestion. We have added the following sentence (Line 236-237): 

Because visual acuity measures central vision, it may not be relevant to postural control.

220 did they test the other eye?

- They also tested the other eye and found no significant results, as stated in Reference 6. We have changed the sentence as follows (Line 241-243): 

For instance, one study reported a correlation between glaucoma severity and equilibrium, noting that pronounced visual field loss in the better eye, but not in the worse eye, correlated with augmented standing sway rates. 

229 “cohort” or “population” as a group, not “participants,” served to “comprise” individuals.

- We have revised the sentence as follows (Line 252-253): 

The cohort in our study predominantly comprised individuals with mild-to-moderate acquired visual deficits.

235 “identified” not “earmarked”

- We have changed the sentence as follows (Line 256-258): 

Four cardinal interventions—education, medical assessment, physical exercise, and environmental modification—have been identified for fall prevention among visually impaired older individuals[31].

259 In 231 you indicated “the influence of visual impairment on fall risk appeared minimal” but here it actually decreases the risk? Are you comparing to effect of vestibular loss?

- We meant that the risk was lower in patients with vestibular disorders. We have changed the sentence as follows (Line 281-283): 

In summary, our findings underscore that most patients with AMD and glaucoma attending an ophthalmology clinic exhibit standard postural control and have a lower falls risk than patients with vestibular disorders.

Reviewer #2

Thank you for your suggestions and comments.

In this study the parameters of fall and equilibre were determined in patients with AMD and glaucoma. The visual function parameters included only visual acuity and visual field. It is necessary to extend the correlations to additional parameters including for AMD low-luminance visual acuity and dark adaptation, and for glaucoma the peripheral visual field limits as determined by Humphrey suprathershold perimetry and Goldmann perimetry.

- In Japan, additional tests, including low-luminance visual acuity and dark adaptation tests, that you have indicated, are not required. The Humphrey Suprathershold Perimetry Program is not available for instruments sold in Japan. Therefore, we did not include these data. 

 Goldmann perimetry, which measures a wide field of vision, is measured at least once, but in clinical practice, it may or may not be measured once every few years; therefore, the data may not reflect the field of vision at the time of registration in this study. Additionally, Goldmann perimetry was nearly normal as most of the present cases were in the early stages of glaucoma.

- We have changed the sentences as follows (Line 107-109):

We also measured the visual fields extensively using Goldmann visual field meters; however, we did not analyze these measurements because they are not routinely performed in clinical practice and may not reflect the visual fields at the time of enrollment in this study.

Reviewer #3

Thank you for your suggestions and comments.

Importantly, I note that the authors have used several complex and peculiar words throughout the manuscript, for example, orchestrated, amassed, afflicted, chronicled, unbeknownst, emanates etc. I think authors have used ChatGPT, which they have acknowledged. It would be better to use simple academic words that are used in scientific writing so that it is easier for all readers to understand the contents.

- This manuscript was proofread by ChatGPT and a native English speaker. Based on Reviewer #3’s suggestion, the entire manuscript was thoroughly proofread again by a native English speaker.

1.The age of patients with AMD and glaucoma is missing in the Abstract. Please add this detail here.

- We have added the median age of each group (Line 21-22).

2.Line 23: While authors have mentioned falls determinants tools but the list of equilibrium function tests are not mentioned.

3.The authors have described 4 falls determinants tools in method sections in the main text. However, Hospital Anxiety and Depression Scale is missed in the Abstract.

- We have changed the sentence as follows (Line 23-28): 

The evaluation metrics included pathological eye movement analysis, bedside head impulse test, single-leg upright test, eye-tracking test, optokinetic nystagmus, and posturography. Furthermore, we administered questionnaires for fall risk determinants, including the Dizziness Handicap Inventory, Activities-Specific Balance Confidence Scale, Falls Efficacy Scale-International, and Hospital Anxiety and Depression Scale. 

4.Line 31: Do visual deficits include both visual acuity (VA) and visual fields? Please specify this for clarity.

- We have changed the sentence as follows (Line 33-36): 

No significant correlation was observed between visual acuity or field deficits and body equilibrium function or fall risk. However, greater peripheral visual field impairment was associated with a tendency for sensory reweighting from visual to somatosensory.

5.The introduction section would benefit by adding some further information about a wide range of falls risk factors and then stating that impaired vision is one of the important falls risk factors. For example, visual risk factors for falls in community-dwelling older adults or fall hazards in the home environment such as inadequate lighting, slippery surfaces, worn carpets, staircases without railings etc. as older adults spent the majority of their time in their homes.

- We have revised the sentence as follows (Line 46-50): 

The risk of falls in older adults is caused by a complex combination of factors. For example, as older adults spend most of their time at home, fall risk factors in the home environment, including inadequate lighting, slippery surfaces, worn carpets, and stairs without handrails, become important. Additionally, visual deficits, which are prevalent in older adults, have been identified as potential risk factors of falls.

6.Line 43 to 44: It seems jumping directly to AMD and glaucoma. It would be better if the authors, in the first stance, mention that AMD and glaucoma cause central vision impairment and peripheral field losses and then describe these two conditions.

- Based on the opinions of Reviewers #3 and #1, we have changed the sentence as follows (Line 69-77): 

In this study, we included patients with age-related macular degeneration (AMD) and glaucoma with moderate or low severity to distinguish between visual acuity impairment and visual field impairment, and determine which of the two contributes to postural control. AMD is a degenerative condition that affects the macula and the central region of the retina, leading to visual acuity impairment. Globally, AMD causes irreversible visual impairment, which is sometimes bilateral and can lead to blindness in severe cases[7]. As the Japanese population ages, the number of AMD cases has surged. Glaucoma, which is characterized by optic nerve damage primarily due to elevated intraocular pressure, results in visual field narrowing and loss. Over 10% of individuals aged ≥60 years in Japan have glaucoma, making it the primary cause of blindness in the country[8].

7.Line 45: Is AMD cause of blindness or cause of irreversible visual impairment? Please check this.

- AMD causes irreversible visual impairment, which is sometimes bilateral and can lead to blindness in severe cases. This has been added to lines 73-74.

8.Line 54: What does visual deficits refer to? Is it reduced vision (VA, contrast sensitivity or visual field) in general or specifically reduced VA and loss of visual fields? Please clarify this.

- We have modified "visual deficits" to "visual acuity” and “field deficits" for clarification. (Line 55)

9.The authors have directly reported demographic findings in the Result section. It is better to include at least the number and age of participants in this section also.

- We have added the number of participants and their median ages to the METHOD section as follows (Line 87-88):

Thirty patients with AMD (median age, 76.0 years) and 30 patients with glaucoma (median age, 64.5 years) participated in this study (Table 1).

10.Line 78: Again, it would be nice if VA and visual fields are stated clearly.

- We have modified the sentence for clarification as follows (Line 85-87):

Patients diagnosed with AMD and glaucoma, representing visual acuity and visual field deficits, respectively, were enrolled from the Ophthalmology Department of our hospital.

11.Visual function test: 

I. What chart was used to test VA and how the process was entailed? Was VA tested monocularly or binocularly and which eye VA was used for correlation analysis in Table 3, is it better eye? I note authors have reported RE and LE in Result section in Table 1 and mentioned in the result section that better eye was used.

- Visual acuity was measured as the decimal visual acuity of the best-corrected visual acuity using a Landolt ring. One eye was shielded, and monocular visual acuity was measured. This has been added to line 97-98. 

The better eye was used for correlation analysis (Table 3). This is noted as "The superior-performing eye’s values were utilized for both visual acuity and visual field evaluations." in Line 103-104.

II. The authors have mentioned that decimal VA was converted into logMAR. I am bit curious why VA was not directly measured and recorded in logMAR.

- In Japan, the routine practice is to measure VA in terms of decimal visual acuity using Landolt rings in Japan. As such, logMAR had to be calculated from the decimal VA results.

III. It is a bit strange that authors have not tested VA in patients with glaucoma. Please justify this.

- Visual acuity data for glaucoma patients were omitted, but we have now included this data because it is important to show that the central vision of patients with glaucoma is preserved (Table 1).

IV. How were visual fields tested monocularly or binocularly and which eye MD was used for correlation analysis, is it again better eye? Please explain these details. The authors have reported MD value for RE only in Table 1.

- The visual fields were tested monocularly, and the better eye was used for the correlation analysis. This was added on line 98-104. 

The values for the left eye were omitted in error; these have now been added.

V. The authors have described how MD values were transmuted to power value, which is new term. Please add references to support this.

- The unit of the MD value is dB or log scale, so it was converted to make the arithmetic calculation reasonable. However, as Reviewer #3 said, this is an unfounded variable conversion; therefore, we have reanalyzed the MD value as it is. We have corrected and removed the relevant descriptions.

12.Outcome measures

I. It would be better to explain equilibrium function test at first and then falls risk evaluation tools later. More detailed explanation is also required for a range of equilibrium function tests.

- We have exchanged the explanation of the fall risk evaluation tools with that of the equilibrium function test. We have also added a detailed explanation of the equilibrium function test as follows (Line 114-126): 

 The HIT is a technique used to evaluate semicircular canal dysfunction by observing the vestibulo–ocular reflex. The examiner sat facing the participants and asked them to look at the tip of the examiner's nose. The examiner then firmly grasped the participant’s temporal region with both hands and applied a fast, small rotation of the head for impulse stimulation. In cases of semicircular canal dysfunction, head impulse stimulation in the direction of the affected side produces a "catch up saccade.” The head impulse was applied three times each to the left and right sides, and a positive result was obtained when the catch-up saccade was observed two or more times[9]. The ETT measures eye movements by fixing the participant’s head and having the eyes tracking a smooth-moving optotype (amplitude, 40°; frequency, 0.3 Hz) in front of the eye. Smooth eye movements were considered normal, and saccadic and ataxic patterns were considered abnormal[10]. The OKN measures eye movement while looking at an object moving at a constant angular velocity in front of the eye. Stimuli of 60°/s were applied in the left and right directions. Saccadic patterns were considered abnormal[11].

II. The tool used by authors contain a set of questionnaires. How were responses from participants obtained? Was it self-reported or face-to-face interview? These all details are required.

- All fall risk evaluations were conducted using self-report questionnaires. This has been added to line 149-150.

13.The authors have not mentioned anything about screening of cognitive function (E.g., Mini-mental state examination) in older adults. I assume some older adults may have cognitive impairments which is likely to have impact on subjective responses that were obtained from questionnaires of falls evaluation tool. Please explain this for further clarification.

14.Line 144 to145: The authors have directly reported results of general health but have not described in method sections that how this information was obtained.

- All patients were current attendees of the hospital, and their medical records were used to investigate their medical history and cognitive function. 

Patients diagnosed with cognitive impairment were excluded because cognitive impairment can affect the results of the questionnaire. 

This was added on line 88-90 & 94-95.

15.In Table 2, I would suggest combining 1, 2 and 3 falls because of the smaller number of participants in each category. There are only 2 participants in each group who had falls within last 6 months. Please run this analysis again.

- We have corrected the "The number of falls within the last 6 months" section in Table 2.

16.Are values reported in the Table 3 Pearson correlations? The interpretation of findings is hard to follow here.

- Spearman’s correlation was used instead of Pearson’s correlation because the amount of data was small and normality was not ensured. This is described in line 158-159.

17.Line 199: Please replace word inferior by less for appropriate word.

- We have changed the sentence as follows (Line 220-221): 

Notably, the proportion of patients with a heightened fall risk was lower than that of patients with peripheral vestibular disorders.

---

## [Decision Letter · Decision Letter 1]

15 Mar 2024

Assessing fall risk and equilibrium function in patients with age-related macular degeneration and glaucoma: An observational study

PONE-D-23-39304R1

Dear Dr. Tokunaga,

We’re pleased to inform you that your manuscript has been judged scientifically suitable for publication and will be formally accepted for publication once it meets all outstanding technical requirements.

Kind regards,

Renato S. Melo, PhD

Academic Editor

PLOS ONE

Additional Editor Comments (optional):

Reviewers' comments:

Reviewer's Responses to Questions

**Comments to the Author**

1. If the authors have adequately addressed your comments raised in a previous round of review and you feel that this manuscript is now acceptable for publication, you may indicate that here to bypass the “Comments to the Author” section, enter your conflict of interest statement in the “Confidential to Editor” section, and submit your "Accept" recommendation.

Reviewer #2: All comments have been addressed

Reviewer #3: (No Response)

2. Is the manuscript technically sound, and do the data support the conclusions?

Reviewer #2: Yes

Reviewer #3: Yes

3. Has the statistical analysis been performed appropriately and rigorously? 

Reviewer #2: Yes

Reviewer #3: No

4. Have the authors made all data underlying the findings in their manuscript fully available?

Reviewer #2: Yes

Reviewer #3: Yes

5. Is the manuscript presented in an intelligible fashion and written in standard English?

Reviewer #2: Yes

Reviewer #3: Yes

6. Review Comments to the Author

Reviewer #2: No further comments all points have been addressed in my previous review and comments I suggest acceptance

Reviewer #3: Dear authors,

Thank you for working and addressing the concerns raised in the last review. The authors have addressed comments satisfactorily. I have no specific comments but a few minor corrections are required:

1. Line 69 to 71:

The authors have added new sentences: ‘In this study, we included patients with age-related macular degeneration (AMD) and glaucoma with moderate or low severity to distinguish between visual acuity impairment and visual field impairment, and determine which of the two contributes to postural control.’ This sentence should go in the Method section rather than in the Introduction section.

2. For one of my previous comments, regarding the reasons for not testing VA in patients with glaucoma (Visual function test: Comment III), the authors have responded that they have now included this data in the revised version. I noted that they have included these data in Table 1, however, they still have not stated in the method section that they assessed visual acuity in patients with glaucoma. Please include this.

3. Line 104 to 106:

The authors have added new sentences: ‘Humans are semi-crossed binocular animals; therefore, they rely on information from the better eye to discriminate objects and control eye movements and postural reflexes derived from peripheral vision.’ Please add references to support this sentence.

7. PLOS authors have the option to publish the peer review history of their article (what does this mean?). If published, this will include your full peer review and any attached files.

Reviewer #2: No

Reviewer #3: No

---

## [Editor Report · Acceptance letter]

21 Mar 2024

PONE-D-23-39304R1 

PLOS ONE

Dear Dr. Tokunaga, 

I'm pleased to inform you that your manuscript has been deemed suitable for publication in PLOS ONE. Congratulations! Your manuscript is now being handed over to our production team.

Kind regards, 

on behalf of

Dr. Renato S. Melo 

Academic Editor

PLOS ONE